METHODS AND RESOURCES

# Sticky Pi is a high-frequency smart trap that enables the study of insect circadian activity under natural conditions

**Quentin Geissmann**[1,2,3]*, **Paul K. Abram**[4], **Di Wu**[3], **Cara H. Haney**[1,2], **Juli Carrillo**[3]

**1** Department of Microbiology and Immunology, The University of British Columbia, Vancouver, British Columbia, Canada, **2** Michael Smith Laboratories, The University of British Columbia, Vancouver, British Columbia, Canada, **3** Faculty of Land and Food Systems, The University of British Columbia, Vancouver (Unceded xʷməθkʷəy̓əm Musqueam Territory), British Columbia, Canada, **4** Agriculture and Agri-Food Canada, Agassiz, British Columbia, Canada

\* qgeissmann@gmail.com

**Data Availability Statement:** All numerical data used to plot relevant figures is documented and available on Figshare (10.6084/m9.figshare. 19764199.v1). All supplementary videos and tables

## Abstract

In the face of severe environmental crises that threaten insect biodiversity, new technologies are imperative to monitor both the identity and ecology of insect species. Traditionally, insect surveys rely on manual collection of traps, which provide abundance data but mask the large intra- and interday variations in insect activity, an important facet of their ecology. Although laboratory studies have shown that circadian processes are central to insects' biological functions, from feeding to reproduction, we lack the high-frequency monitoring tools to study insect circadian biology in the field. To address these issues, we developed the Sticky Pi, a novel, autonomous, open-source, insect trap that acquires images of sticky cards every 20 minutes. Using custom deep learning algorithms, we automatically and accurately scored where, when, and which insects were captured. First, we validated our device in controlled laboratory conditions with a classic chronobiological model organism, *Drosophila melanogaster*. Then, we deployed an array of Sticky Pis to the field to characterise the daily activity of an agricultural pest, *Drosophila suzukii*, and its parasitoid wasps. Finally, we demonstrate the wide scope of our smart trap by describing the sympatric arrangement of insect temporal niches in a community, without targeting particular taxa a priori. Together, the automatic identification and high sampling rate of our tool provide biologists with unique data that impacts research far beyond chronobiology, with applications to biodiversity monitoring and pest control as well as fundamental implications for phenology, behavioural ecology, and ecophysiology. We released the Sticky Pi project as an open community resource on https://doc.sticky-pi.com.

## Introduction

In order to fully characterise ecological communities, we must go beyond mere species inventories and integrate functional aspects such as interspecific interactions and organisms' behaviours through space and time [1,2]. Chronobiology, the study of biological rhythms, has

are on hosted on figshare. Here is their DOIs: S1 Table: 10.6084/m9.figshare.15135819 S2 Table: 10.6084/m9.figshare.15135825 S3 Table: 10.6084/m9.figshare.19653357 S1 Video: 10.6084/m9.figshare.15135702 S2 Video: 10.6084/m9.figshare.15135714 S3 Video: 10.6084/m9.figshare.15135735 S4 Video: 10.6084/m9.figshare.15135744 S5 Video: 10.6084/m9.figshare.15135750.

**Funding:** Q.G. was funded by the International Human Frontier Science Program Organization (LT000325/2019). This research (funding to P.K.A., C.H.H. and J.C.) is part of Organic Science Cluster 3, led by the Organic Federation of Canada in collaboration with the Organic Agriculture Centre of Canada at Dalhousie University, supported by Agriculture and Agri-Food Canada's Canadian Agricultural Partnership - AgriScience Program. P. K.A. was supported by funding from Agriculture and Agri-Food Canada. This work was also supported by a Seeding Food Innovation grant from George Weston Ltd. to C.H.H. and J.C., and a Canada Research Chair award to C.H.H. The funders had no role in study design, data collection and analysis, decision to publish, or preparation of the manuscript.

**Competing interests:** The authors have declared that no competing interests exist.

**Abbreviations:** API, Application Programming Interface; BIN, barcode index number; CO1, cytochrome c oxidase subunit I; MDS, multidimensional scaling; ResNet, Residual Neural Network; SIM, Siamese Insect Matcher; WZT, Warped Zeitgeber time; ZT, Zeitgeber time.

shown that circadian (i.e., internal) clocks play ubiquitous and pivotal physiological roles, and that the daily timing of most behaviours matters enormously [3]. Therefore, understanding not only which species are present, but also when they are active adds a crucial, functional, layer to community ecology.

The emerging field of chronoecology has begun to integrate chronobiological and ecological questions to reveal important phenomena [4,5]. For example, certain prey can respond to predators by altering their diel activity [6], parasites may manipulate their host's clock to increase their transmission [7], foraging behaviours are guided by the circadian clock [8], and, over evolutionary timescales, differences in diel activities may drive speciation [9]. However, because nearly all studies to date have been conducted on isolated individuals in laboratory microcosms, the ecological and evolutionary implications of circadian clocks in natural environments remain largely unknown [10].

While chronobiology requires a physiological and behavioural time scale (i.e., seconds to hours), insect surveys have primarily focused on the phenological scale (i.e., days to months). Compared to bird and mammal studies, where methodological breakthroughs in animal tracking devices have enabled the ecological study of the timing of behaviours, similar tools for invertebrates are lacking [11] or limited to specific cases [12–14]. Promisingly, portable electronics and machine learning are beginning to reach insect ecology and monitoring [15]. For example, "smart traps can now automatise traditional insect capture and identification" [16]. In particular, camera-based traps can passively monitor insects and use deep learning to identify multiple species. However, such tools are often designed for applications on a single focal species and, due to the large amount of data they generate as well as the complexity of the downstream analysis, camera-based traps have typically been limited to daily monitoring and have not previously been used to study insect circadian behaviours.

Here, we present and validate the Sticky Pi, an open-source generalist automatic trap to study insect chronobiology in the field. Our unique framework both automatises insect surveying and adds a novel temporal and behavioural dimension to the study of biodiversity. This work paves the way for insect community chronoecology: the organisation, interaction, and diversity of organisms' biological rhythms within an ecological community.

## Results

### Sticky Pi device and platform

We built the Sticky Pi (Fig 1A–1C), a device that captures insects on a sticky card and images them every 20 minutes. Compared to other methods, our device acquires high-quality images at high frequency, hence providing a fine temporal resolution on insect captures. Devices are equipped with a temperature and humidity sensor and have 2 weeks of autonomy (without solar panels). Sticky Pis are open source, 3D printed, and inexpensive (<200 USD). Sticky Pis can be fitted with cages to prevent small vertebrates from predating trapped insects. Another unique feature is their camera-triggered backlit flashlight, which enhances the contrast, reduces glare, and allows for nighttime imaging. Most sticky cards available on the market are thin and translucent, which allows for the transmission of light. White light was chosen for its versatility: Sticky cards with different absorption spectra can be used. For outdoor use, the camera's built-in infrared-cut filter was not removed. Such filters, which remove infrared light, are standard in photography as they reduce chromatic aberrations. As a result, we can discern 3 mm-long insects on a total visible surface of 215 cm$^2$ (Fig 1D and 1E), which is sufficient to identify many taxa. In order to centralise, analyse, and visualise the data from multiple devices, we developed a scalable platform (S1 Fig), which includes a suite of services: an Application Programming Interface (API), a database, and an interactive web application (S1 Video).

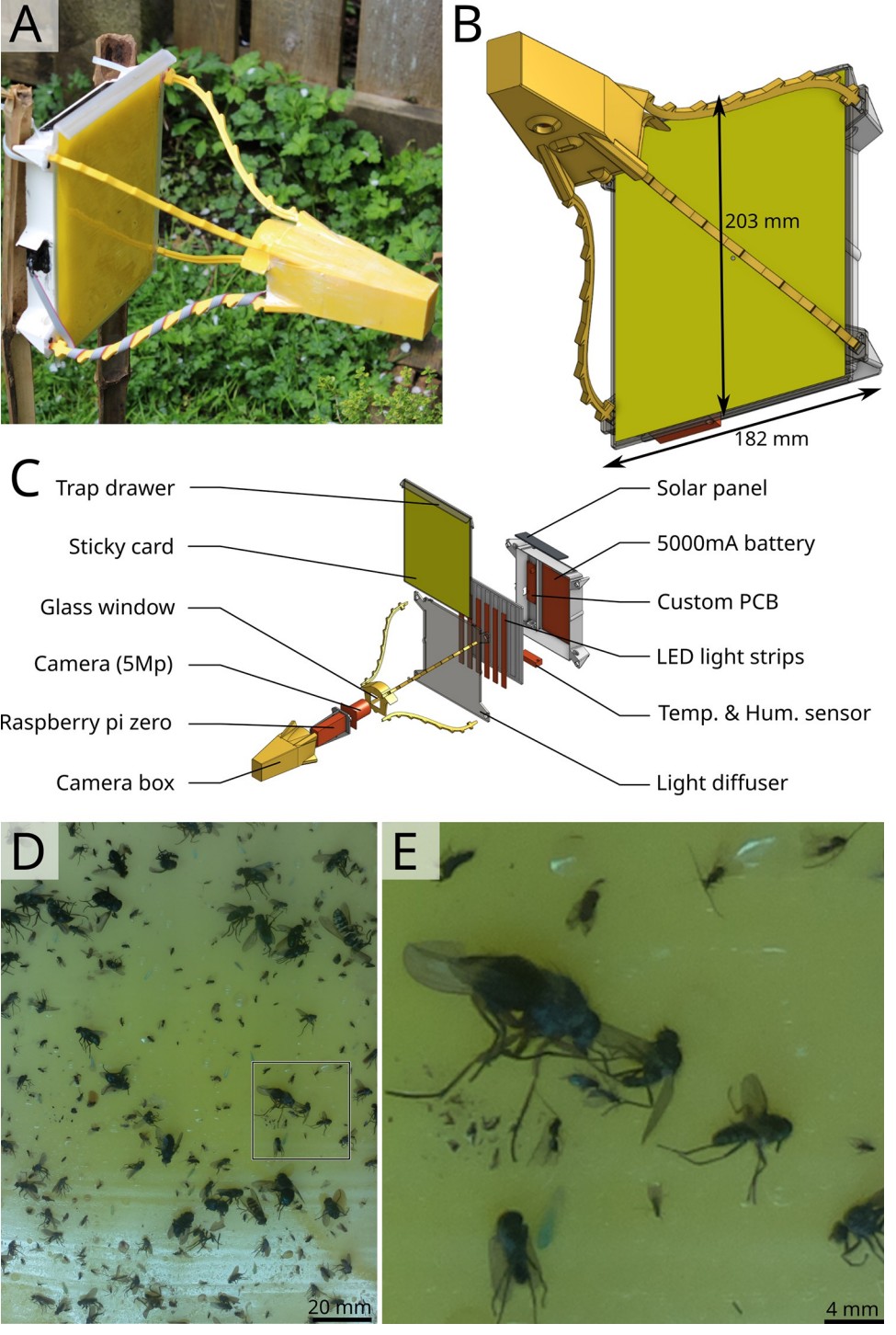

**Fig 1. Sticky Pi device. (A, B)** Assembled Sticky Pi. The device dimensions are 326×203×182 mm (*d*×*w*×*h*). **(C)** Exploded view, showing the main hardware components. Devices are open source, affordable, and can be built with off-the-shelf electronics and a 3D printer. Each Sticky Pi takes an image every 20 minutes using an LED backlit flash. **(D)** Full-scale image as acquired by a Sticky Pi (originally 1944×2592 px, 126×126 mm). **(E)** Magnification of the 500×500 px region shown in D.

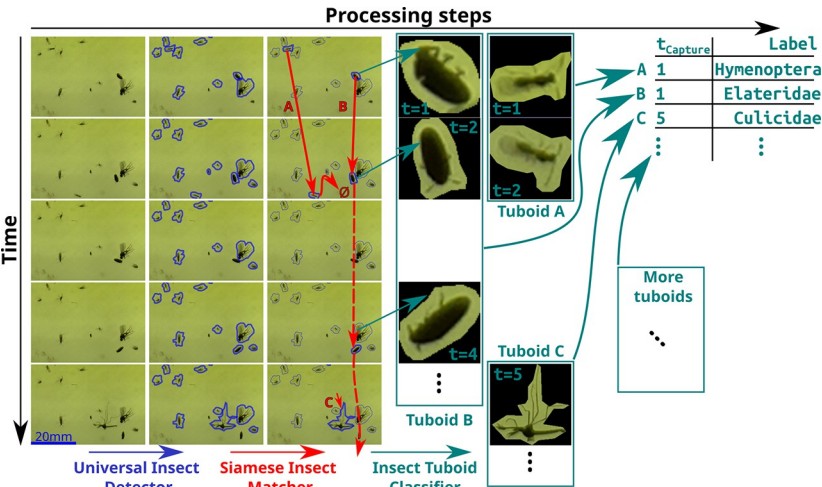

**Fig 2. Overview of the image processing workflow.** Devices acquire images approximately every 20 minutes, which results in a 500 image-long series per week per device. Rows in the figure represent consecutive images in a series. Series are analysed in 3 main algorithms (left to right). First, the Universal Insect Detector applies a 2-class Mask R-CNN to segment insect instances (versus background), blue. Second, the SIM applies a custom Siamese network–based algorithm to track instances throughout the series (red arrows), which results in multiple frames for the same insect instance, i.e., "insect tuboids. Last, the Insect Tuboid Classifier uses an enhanced ResNet50 architecture to predict insect taxonomy from multiple photographs. SIM, Siamese Insect Matcher.

Deployment and maintenance instructions are detailed in our documentation (https://doc. sticky-pi.com/web-server.html).

## Image processing

In order to classify captured insects, we developed a novel analysis pipeline, which we validated on a combination of still photographs of standard sticky traps and series of images from 10 Sticky Pis deployed in 2 berry fields for 11 weeks (see Methods section and next result sections). We noticed trapped insects often move, escape, are predated, become transiently occluded, or otherwise decay (S2 Video). Therefore, we used cross-frame information rather than independently segmenting and classifying insects frame by frame. Our pipeline operates in 3 steps (summarised below and in Fig 2): (i) the Universal Insect Detector segments insect instances in independent images assuming a 2-class problem: insect versus background; (ii) the Siamese Insect Matcher (SIM) tracks insect instances between frames, using visual similarity and displacement; and (iii) The Insect Tuboid Classifier uses information from multiple frames to make a single taxonomic prediction on each tracked insect instance.

## Universal insect detector

To segment "insects from their "background, we based the Universal Insect Detector on Mask R-CNN [17] and trained it on 240 hand-annotated images from Sticky Pis as well as 120 "foreign" images acquired with different devices (see Methods section). On the validation dataset, our algorithm had an overall 82.9% recall and 91.2% precision (S2 Fig). Noticeably, recall increased to 90.5% when excluding the 25% smallest objects (area < 1,000 px. i.e., 2.12 mm2), indicating that the smallest insect instances are ambiguous. When performing validation on the foreign dataset of 20 images acquired with the Raspberry Pi camera HQ, we obtained a precision 96.4% and a recall of 92.2%, indicating that newly available optics may largely increase segmentation performance (although all experimental data in this study were obtained with the original camera, before the HQ became available).

## Siamese insect matcher

In order to track insects through multiple frames, we built a directed graph for each series of images; connecting instances on the basis of a matching metric, which we computed using a custom Siamese Neural Network (S3A Fig and Methods section). We used this metric to track insects in a 3-pass process (S3B Fig and Methods section). This step resulted in multiframe representations of insects through their respective series, which we call "tuboids." S3 Video shows a time-lapse video of a series where each insect tuboid is boxed and labelled with a unique number.

## Insect Tuboid classifier

To classify multiframe insect representations (i.e., tuboids), we based the Insect Tuboid Classifier (Fig 3), on a Residual Neural Network (ResNet) architecture [18] with 2 important modifications: (i) We explicitly included the size of the putative insect as an input variable to the fully connected layer as size may be important for classification and our images have consistent scale; and (ii) Since tuboid frames provide nonredundant information for classification (stuck insects often still move and illumination changes), we applied the convolution layers on 6 frames sampled in the first 24 h and combined their outputs in a single prediction (Fig 3A). In this study, we trained our classifier on a dataset of 2,896 insect tuboids, trapped in 2 berry fields in the same location and season (see next result sections and Methods section). We defined 18 taxonomic labels, described in S1 Table, using a combination of visual identification and DNA barcoding of insects sampled from the traps after they were collected from the field (S2 Table and Methods sections). Fig 3B and 3C shows representative insect images corresponding to these 18 labels (i.e., only 1 frame from a whole multiframe tuboid) and summary statistics on the validation dataset (982 tuboids). S3 Table present the whole confusion matrix for the 18 labels.

The overall accuracy (i.e., the proportion of correct predictions) is 78.4%. Our dataset contained a large proportion of either "Background objects" and "Undefined insects" (16.2% and 22.4%, respectively). When merging these 2 less informative labels, we reach an overall 83.1% accuracy on the remaining 17 classes. Precision (i.e., the proportion of correct predictions given a predicted label) and recall (i.e., the proportion of correct prediction given an actual label) were high for the Typhlocybinae (leafhoppers) morphospecies (92% and 94%). For *Drosophila suzukii* [Diptera: Drosophilidae] (spotted-wing drosophila), an important berry pest, we labelled males as a separate class due to their distinctive dark spots and also reached a high precision (86%) and recall (91%)—see detail in S3 Table. These results show that performance can be high for small, but abundant and visually distinct taxa.

## Sticky Pis can quantify circadian activity in laboratory conditions

To test whether capture rate on a sticky card could describe the circadian activity of an insect population, we conducted a laboratory experiment on vinegar flies, *Drosophila melanogaster* [Diptera: Drosophilidae], either in constant light (LL) or constant dark (DD), both compared to control populations held in 12:12 h Light:Dark cycles (LD) (Fig 4). From the extensive literature on *D. melanogaster*, we predicted a crepuscular activity LD and DD (flies are free-running in DD), but no rhythm in LL [19]. We placed groups of flies in a large cage that contained a single Sticky Pi (simplified for the laboratory and using infrared light; Methods section). The DD and LL experiments were performed independently and each compared to their own internal LD control. The use of a infrared optics and lighting resulted in lower quality images (i.e., reduced sharpness). However, in this simplified scenario, there were no occlusions, and the classification was binary (fly versus background). Therefore, we used a direct approach:

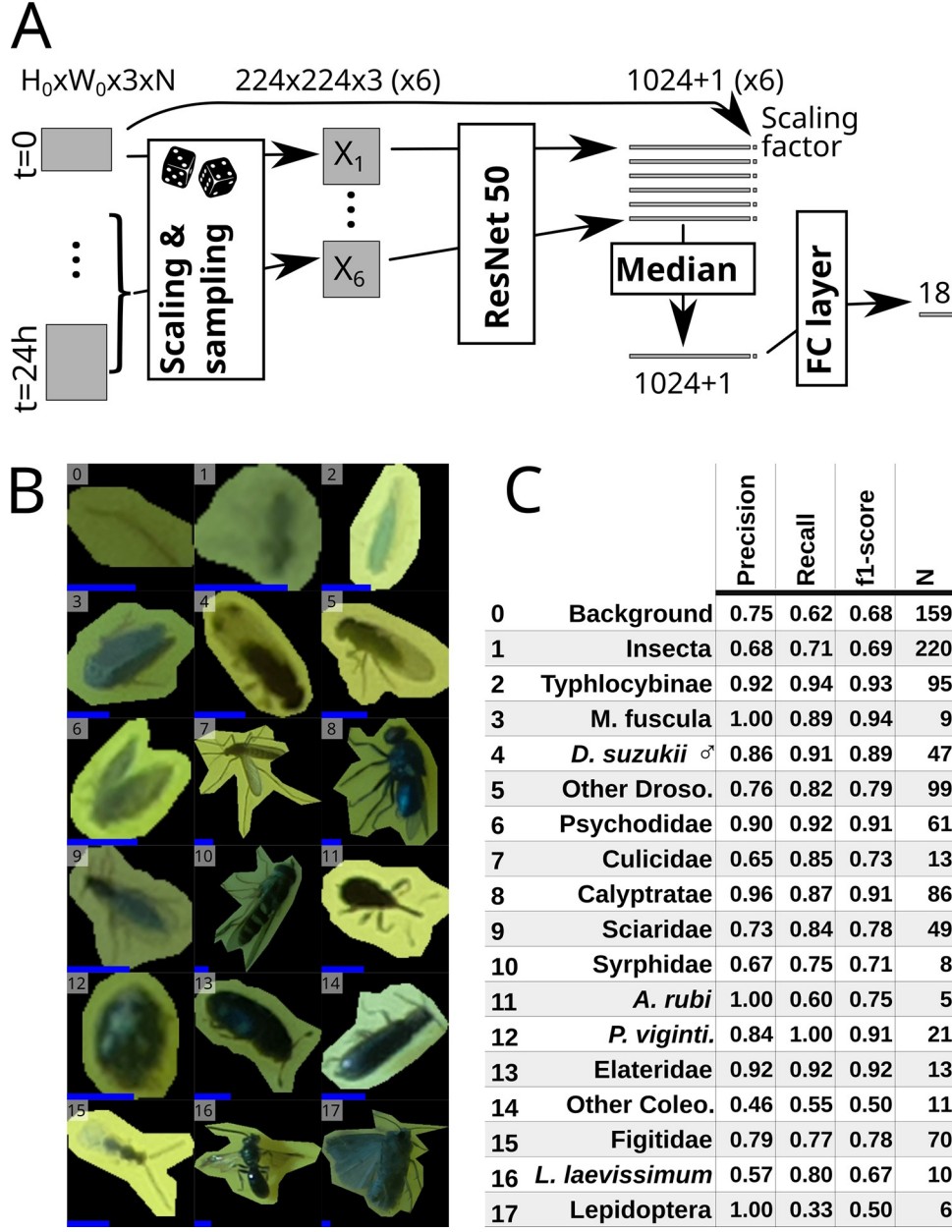

**Fig 3. Insect Tuboid Classifier description and performance.** (**A**) Algorithm to classify insect tuboids. The first image as well as 5 randomly selected within the first day of data are selected. Each image is scaled and processed by a ResNet50 network to generate an output feature vector per frame. Each vector is augmented with the original scale of the object, and the element-wise median over the 6 frames is computed. The resulting average feature vector is processed by a last, fully connected, layer with an output of 18 labels. (**B**) Representative examples of the 18 different classes. Note that we show only 1 image, but input tuboids have multiple frames. All images were rescaled and padded to 224×224 px squares: the input dimensions for the ResNet50. The added blue scale bar, on the bottom left of each tile, represents a length of 2 mm (i.e., 31 px). (**C**) Classification performance, showing precision, recall and f1-score (the harmonic mean of the precision and recall) for each label. Row numbers match labels in B. See S3 Table for the full confusion matrix. Abbreviated rows in C are *Macropsis fuscula* (3), *Drosophila suzukii* males (4), drosophilids that are not male *D. suzukii* (5), *Anthonomus rubi* (11), *Psyllobora vigintimaculata* (12), Coleoptera that do not belong to any above subgroup (14), and *Lasioglossum laevissimum* (16).

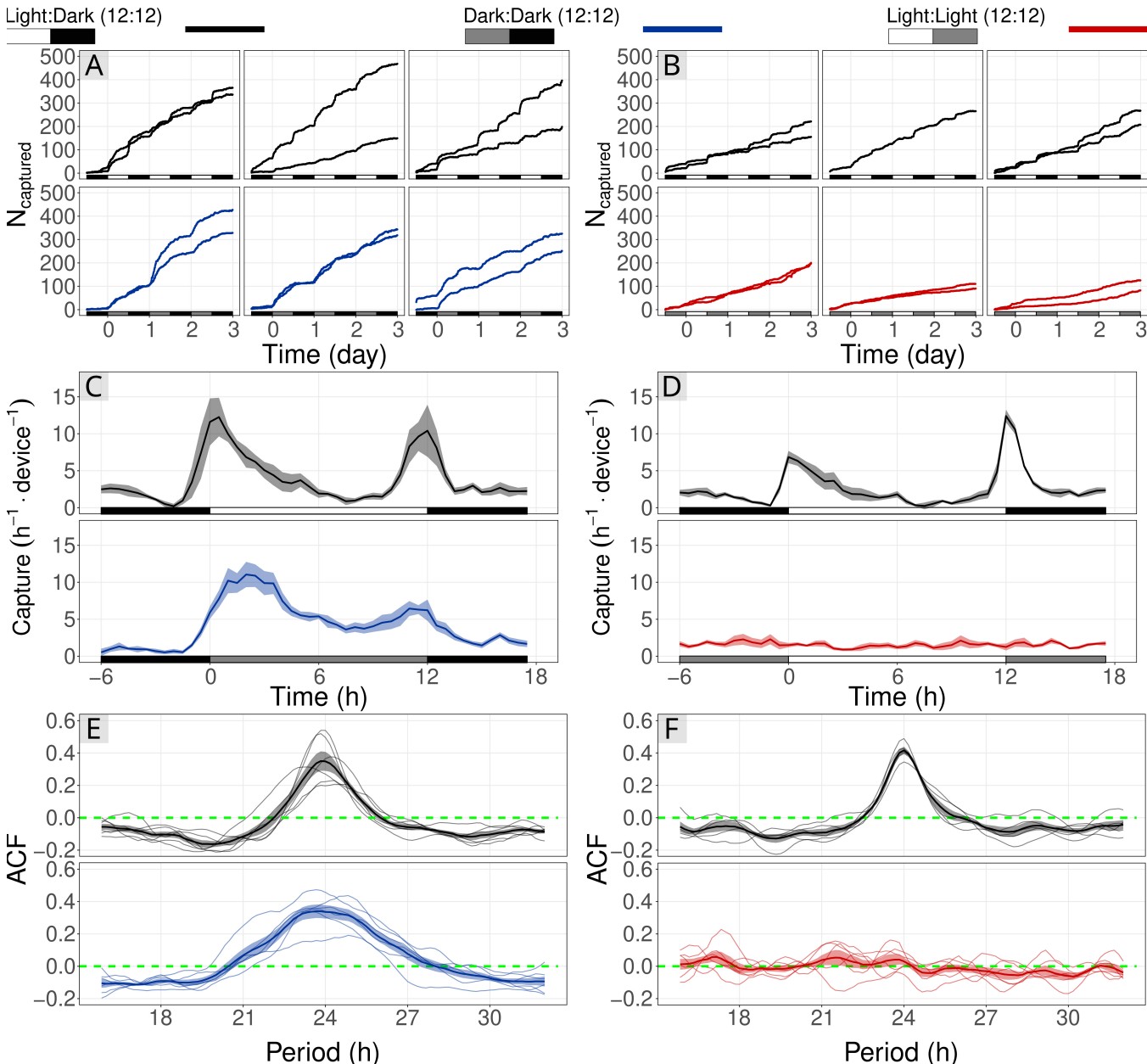

**Fig 4. Sticky Pis can monitor circadian rhythms of free-moving populations in the laboratory.** Vinegar flies, ***Drosophila melanogaster***, were held in a large cage with a Sticky Pi. We conducted 2 experiments to show the effect of Light:Light (red; A, C, E) and Dark:Dark (blue; B, D, F) light-regimes on capture rate. Each was compared to a control population that remained in the entrainment conditions: Light:Dark, 12:12 h cycles (black). **(A, B)** Cumulative number of insects captured over time. Columns of the panels correspond to independent full replicates. We used 2 devices per condition, in each full replicate. **(C, D)** Capture rates over circadian time. As expected, capture rates in LD and DD show a clear crepuscular activity, but no activity peak in constant light. **(E, F)** Autocorrelation of capture rates. Each thin line represents a series (i.e., one device in one full replicate), and the thick line is the average autocorrelogram. The green dotted line shows the expectation under the hypothesis that there is no periodic pattern in capture rate (ACF). The underlying data for this figure can be found on figshare [20]. ACF, AutoCorrelation Function.

We trained and applied independent Mask-RCNN to segment flies from their background. Then, rather than tracking insects (using our SIM), we extracted the raw counts from each frames and applied a low-pass filter (see Methods section). Consistent with previous studies on circadian behaviour of *D. melanogaster*, populations in both LD and DD conditions exhibited

strong rhythmic capture rates, with an approximate period of 24 h: 23.8 h and 23.67 h, respectively. For example, their overall capture rate was approximately 0.6 h$^{-1}$ between ZT22 and ZT23 h, but peaked at 9.5 h$^{-1}$ between ZT01 and ZT02 h (Fig 4C and 4E). The average autocorrelation (measure of rhythmicity), with a 24 h lag, for the both DD populations and their internal control were high and significant: 0.34 (p−value<10$^{-3}$, $N = 6$, $t$ test) and 0.35 (p−value = 2×10$^{-3}$, $N = 6$, $t$ test), respectively.

Also as hypothesised, the fly populations held in constant light (LL) showed no detectable behavioural rhythm and had a constant average capture rate of 1.6 h$^{-1}$ (SD = 0.62) (Fig 4D and 4F). The average autocorrelation with a 24 h lag for the 6 LL series was 0.03 and was not significantly different from zero (p−value>0.24, $t$ test), which shows the absence of detectable 24-h rhythm. In contrast, the 5 series of the LD internal control had a large and significant autocorrelation value of 0.42 (p−value<10$^{-4}$, $t$ test). Collectively, these observations indicate that Sticky Pis have the potential to capture circadian behaviour in a free-flying insect population.

## Sticky Pis quantify activity rhythms of wild *Drosophila suzukii*

To test the potential of the Sticky Pis to monitor wild populations of free-moving insects in the field, we deployed 10 traps in a blackberry field inhabited by the well-studied and important pest species *D. suzukii* (see Methods section). Like *D. melanogaster*, *D. suzukii* has been characterised as crepuscular both in the laboratory [21] and, with manual observations, in the field [22]. Since capture rates can be very low without attractants [22], we baited half (5) of our traps with apple cider vinegar (see Methods section). In addition to *D. suzukii*, we wanted to simultaneously describe the activity of lesser-known species in the same community. In particular, *D. suzukii* and other closely related Drosophila are attacked by parasitoid wasps [Hymenoptera: Figitidae], 2 of which (*Leptopilina japonica* and *Ganaspis brasiliensis*) have recently arrived in our study region [23]. Their diel activity has not yet been described. In Fig 5, we show the capture rate of male *D. suzukii*, other putative Drosophilidae and parasitoid wasps throughout the 7-week trial (Fig 5A) and throughout an average day (Fig 5B).

Our results corroborate a distinctive crepuscular activity pattern for male *D. suzukii* and other putative drosophilids. For example, 68.0% ($CI_{95\%}$ = [63.9, 71.3], 10,000 bootstrap replicates) of *D. suzukii* and 57.8% ($CI_{95\%}$ = [53.2, 61.6], 10,000 bootstrap replicates) of the other Drosophilids. occurred either in the 4 hours around dawn ($WZT \in [8, 14]$ h) or dusk ($WZT < 2$ or $WZT > 22$ h)under a time-uniform capture null hypothesis, we would expect only $\frac{1}{3}$ of captures in these 8 hours. In contrast, Figitidae wasps were exclusively diurnal, with 83.0% $CI_{95\%}$ = [79.9,85.4], 10,000 bootstrap replicates) of all the captures occurring during the day ($WZT < 12$), where we would expect only 50% by chance.

Overall, baiting widely increased the number of male *D. suzukii* (from 3.0 to 26.0 device$^{-1}$. week$^{-1}$, p−value<2×10$^{-8}$), and other Drosophilidae (from 8.8 to 49.8 device$^{-1}$. week$^{-1}$, p−value<10$^{-9}$), but not parasitoid wasps (p−value>0.65, Wilcoxon rank-sum tests). These findings indicate that Sticky Pi can quantify the circadian behaviour of a simple insect community in a natural setting.

## Sticky Pi, a resource for community chronoecology

Berry fields are inhabited by a variety of insects for which we aimed to capture proof-of-concept community chronoecological data. In a separate trial, we placed 10 Sticky Pis in a raspberry field and monitored the average daily capture rate of 8 selected taxa (Fig 6A) over 4 weeks—we selected these 8 taxa based on the number of individuals, performance of the classifier (Fig 3), and taxonomic distinctness (S4 Fig shows the other classified taxa). We then defined a dissimilarity score and applied multidimensional scaling (MDS) to represent temporal niche proximity

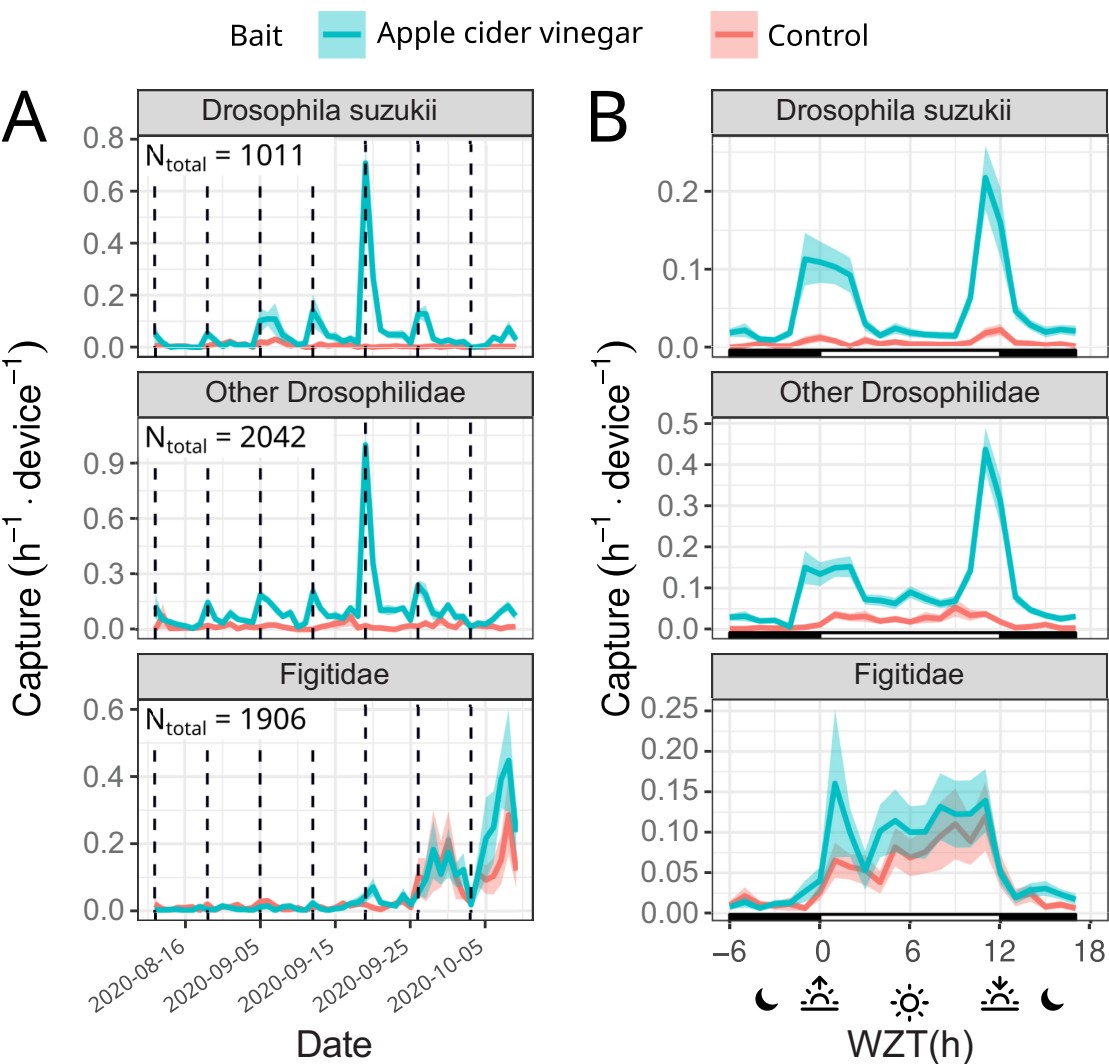

**Fig 5. Sticky Pis can monitor spotted-wing drosophila diel activity in the field.** We deployed 10 Sticky Pis in a blackberry field for 7 weeks and attached apple cider vinegar baits to half of them (blue versus red for unbaited control). This figure shows specifically the males ***Drosophila suzukii***, the other Drosophilidae flies, and the Figitidae wasps. **(A)** Capture rate over time, averaged per day, showing the seasonal occurrence of insect populations. **(B)** Average capture rate over time of the day (note that time was transformed to compensate for changes in day length and onset—i.e., Warped ZT: 0 h and 12 h represent the sunset and sunrise, respectively, see Methods section). Both males *D. suzukii* and the other drosophilids were trapped predominantly on the baited devices. Both populations exhibit a crepuscular activity. In contrast, Figitidae wasps have a diurnal activity pattern and are unaffected by the bait. Error bars show standard errors across replicates (device×week). The underlying data for this figure can be found on figshare [20]. ZT, Zeitgeber time.

in 2 dimensions (see Methods section). We show that multiple taxa can be monitored simultaneously, and statistically partitioned according to their temporal niche (Fig 6B). Specifically, as shown in Fig 6A, sweat bees (*Lasioglossum laevissimum*), large flies (Calyptratae), and hoverflies (Syrphidae) show a clear diurnal activity pattern with a capture peak at solar noon (Warped Zeitgeber time [WZT] = 6h, see Methods section for WZT). Sciaridae gnats were also diurnal, but their capture rate was skewed towards the afternoon, with a peak around WZT = 7 h. The Typhlocybinae leafhopper was vespertine, with a single sharp activity peak at sunset (WZT = 11 h). The Psychodidae were crepuscular, exhibiting 2 peaks of activity, at dusk and dawn. Both mosquitoes (Culicidae) and moths (Lepidoptera) were nocturnal.

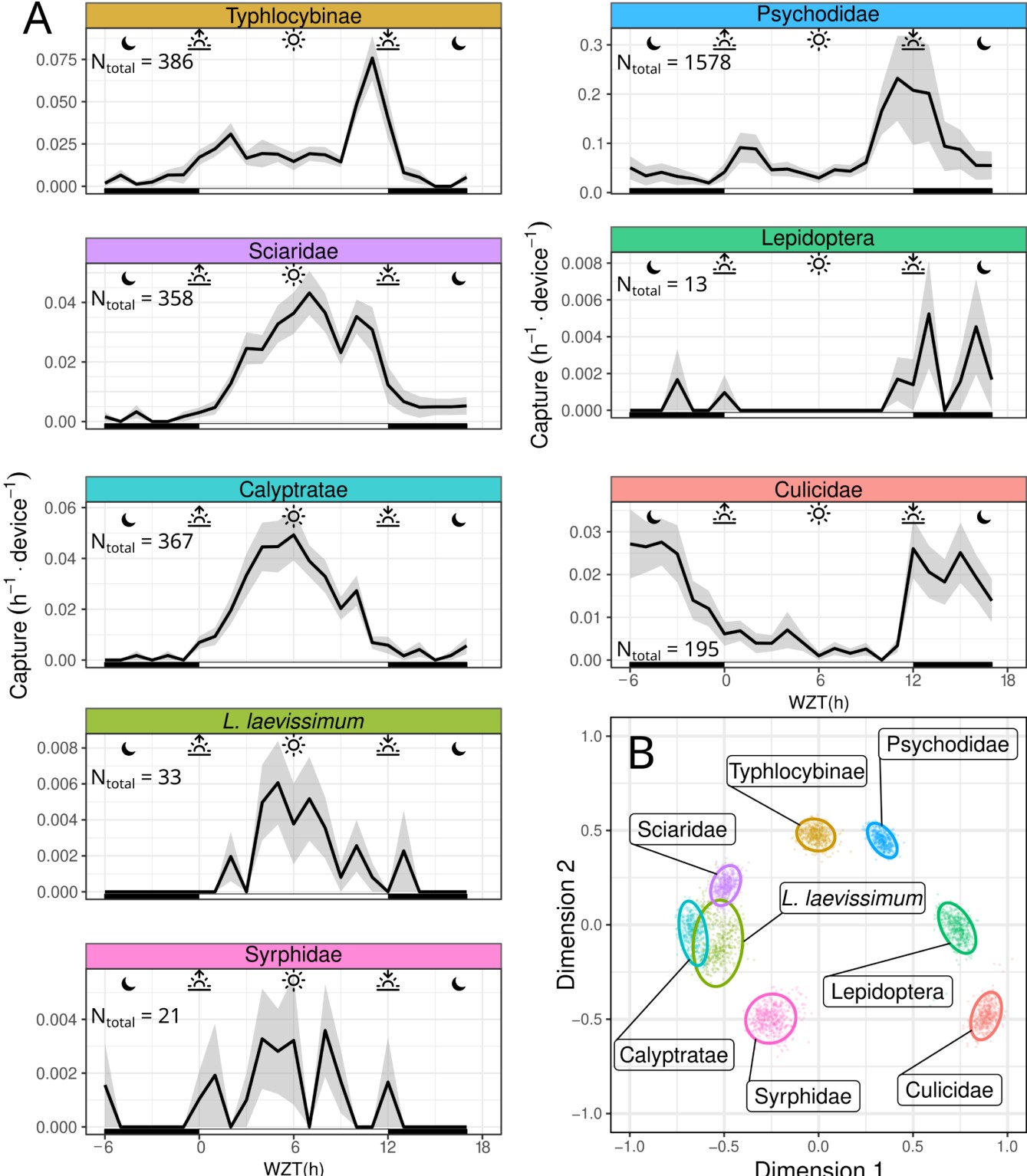

**Fig 6. Sticky Pi reveals community chronoecology.** In order to assess the activity pattern of a diverse community, we deployed 10 Sticky Pis in a raspberry field for 4 weeks, replacing sticky cards weekly. This figure shows a subset of abundant insect taxa that were detected by our algorithm with high precision (see Supporting information for the full dataset). **(A)** Average capture rate over time of the day (note that time was transformed to compensate for changes in day length and onset—i.e., Warped ZT: 0 h and 12 h represent the sunset and sunrise, respectively. See Methods section). **(B)** MDS of the populations shown in A. Similarity is based on the Pearson correlation between the average hourly activity of any 2 populations. Small points are individual bootstrap replicates, and

ellipses are 95% confidence intervals (see Methods section). Insect taxa partition according to their temporal activity pattern (e.g., nocturnal, diurnal, or crepuscular). Error bars in A show standard errors across replicates (device×week). Individual facets in A were manually laid out to match the topology of B. The underlying data for this figure can be found on figshare [20]. MDS, multidimensional scaling; ZT, Zeitgeber time.

We asked to what extent the presence of previous insects on a trap impacted its subsequent capture (e.g., if trap became saturated by insects). We first observed that the cumulative number of all insects on all traps, did not appear to change over time (S6A Fig). Then, to statistically address this question for individual taxa, we reasoned that if the capture rate was linear, the number of insects captured in the first 3 days should be 50% of total (6 full days) $N_{[0,3]days}/N_{[0,6]days} = 1/2$. Thus, we tested whether the final, total, number of insects (from all taxa) explained the proportion of captured insects $N_{[0,3]days}/N_{[0,6]days}$ (of a given taxa) in the first half of the experiment. We found that the number of insects captured in the first half of each trial was not different from 1/2 (intercept) and that the number of insects captured did not explain taxa's capture rate (slope) (S6B Fig, linear models, p−value>0.05 ∀ taxa, *t* tests on model coefficients).

Together, our findings show that, even without a priori knowledge of the diel activity of specific taxa, Sticky Pis can inform about both the community structure and temporal patterns of behaviour of a natural insect community.

## Discussion

We have developed the Sticky Pi, a generalist and versatile insect smart trap that is open source, documented, and affordable (Fig 1). Uniquely, Sticky Pis acquires frequent images to finely describe when specific insects are captured. Since the main limitation to insect chronoecology is the lack of high-frequency population monitoring technologies [11], our innovation promises to spark discoveries at the frontier between 2 important domains: chronobiology and biodiversity monitoring. Furthermore, taking multiple images of the same specimen may improve classification performance. To adapt our tool to big data problems, we designed a suite of web services (S1 Fig) that supports multiple concurrent users, can communicate with distributed resources and may interoperate with other biodiversity monitoring projects and community science platforms [24]. Compared to other camera-based automatic insect traps [25], we opted for a decentralised solution. Our platform is explicitly designed to handle multiple concurrent traps: with unique identifiers for devices and images (and relevant metadata), an efficient mean of retrieving data wirelessly, and a dedicated database, with an API to store, query, and analyse the results. These features, together with a low device cost (<200 USD) will facilitate scaling to the landscape level.

Our device's main limitation is image quality (Fig 1D and 1E). Indeed, high-performance segmentation and classification of insects were limited to specimens larger than 3 millimetres (S2 Fig), hence reducing the taxonomic resolution for small insects. We found that segmentation was globally very precise (>90%) and sensitive (recall > 90% for objects larger than 2 mm$^2$). Furthermore, our machine learning pipeline (Fig 2) showed a high overall accuracy of the Insect Tuboid Classifier (83.1% on average, when merging background and undefined insects; see Fig 3). Camera technology is quickly improving and our segmentation results with the new Raspberry Pi camera HQ (12.3 Mpx, CS mount) are promising, with preliminary results showing both precision and recall greater than 90% overall, on the segmentation task. Some of the inaccuracy in segmentation results from transient occlusion or changes in the image quality. Therefore, tracking (using the SIM) likely improves recall as insects that are missed on some frames may be detected on previous or subsequent frames.

Another potentially limiting feature of our device is the frequency of the images taken (every 20 minutes). According to their context and questions, users can programme the hardware timer with a different interval. However, we judged 3 times per hour a good compromise between time resolution—an hourly resolution being necessary to study chronobiology or the impact of fast weather variations—and battery and data storage efficiency. Furthermore, a more frequent use of the flash light (e.g., every minute) may be more of a disturbance to wildlife [26].

In this respect, our image time lapse approach contrasts with continuous lighter-weight systems such as sensor-based traps, which are suited for high-frequency sampling and were recently employed to study diel activity of a single species [13,27]. However, sensor-based traps are often limited to scenarios with a priori-targeted species that respond to certain specific olfactory or visual baits—which considerably narrows their applicability [28]. In contrast, camera-based traps are more generalist as they can passively monitor insects, using machine learning to identify multiple species. Our system importantly captures and keeps individual insect specimens. While this destructive process comes with limitations, it is also essential in naive contexts, where we do not know a priori which insects may be present. Indeed, physical specimens are needed for visual or DNA-based taxonomic characterisation, in particular when working on diverse or undescribed communities [29]. Keeping individual insects would not be possible if animals were released or kept in a common container. Furthermore, trapping insects permanently greatly reduced the risk of recapturing the same individuals several times.

An important consideration using sticky card is their potential to become saturated with insects. In our study, we replaced traps weekly to limit this possibility. Furthermore, we found no statistical effect of the number of insects on the probability of capture for a given taxa (S6 Fig). However, we advise users to replace sticky cards often enough to limit this risk.

We corroborated circadian results that had historically been obtained on individually housed insects, using heterogeneous populations in large flight cages (Fig 4). This suggests that Sticky Pis could be an alternative tool for laboratory experiments on mixed populations of interacting insects. In the field, we monitored both the seasonal and diel activity of a well-studied pest species: spotted-wing drosophila (*D. suzukii*). Like others before [22], we concluded that wild *D. suzukii* was crepuscular (Fig 5). In the process, we also found that Figitidae wasps —natural enemies of *D. suzukii*—were distinctly diurnal and were most often detected later in the season. Finally, we characterised the diel activity of the flying insect community in a raspberry field, without targeting taxa a priori (Fig 6). With only 10 devices, over 4 weeks, we were able to reveal the diversity of temporal niches, showing coexisting insects with a wide spectrum of diel activity patterns—e.g., diurnal, crepuscular versus nocturnal; bimodal versus unimodal. An additional and noteworthy advantage of time-lapse photography is the incidental description of unexpected behaviours such as insect predation (S4 Video) and the acquisition of specimens that eventually escape from traps (S5 Video).

Importantly, any insect trap only gives a biased estimate of the number and activity on insects at a given time. For sticky cards, the probability of capturing an insect depends, at a minimum, on the flying activity, the size of the population and the trap attractiveness (and the "conversion rate"—i.e., the probability of trapping an insect given it is attracted). Importantly, Sticky Pis only capture mobile adult insects, and, therefore, cannot explicitly quantify the timing of important behaviours such as mating, feeding, egg laying, emergence, and quiescence. However, in many species, locomotion and dispersal are a prerequisite to other activities. Therefore, capture rate implicitly encapsulates a larger portion of the behavioural spectrum. In most cases, the daily variation of population size is likely negligible. In contrast, several factors may render trap attractiveness and conversion rates variable during the day. First, visual cues —which impact a trap's capture rates [30,31]—vary for any capture substrate (e.g., light

intensity and spectral qualities inevitably fluctuate). Second, insects' ability to detect, avoid, or escape traps may temporarily differ. Last, the preference for certain trap features could itself be a circadian trait. For example, the responses of certain insects to colours [32] allelochemicals [33,34] and semiochemicals [35–37] is time modulated. While inconsistencies in trap attractiveness may, in some cases, narrow the scope of the conclusions that can be made with our tool, it also paves the way for research on the diel time budget of many insects. Indeed, studying the contrast in trapping rates between different, ecologically relevant, trap features (e.g., baits, colour, and location) could help to develop new and improved trapping methodologies while bridging chronobiology and behavioural ecology.

In addition to insect capture rates, Sticky Pis also monitor humidity and temperature, which are both crucial for most insects behaviour and demography [38,39]. This study was performed at the level of a small agricultural field, where the local spatial abiotic variations were small compared to the interday variations, and hourly temperatures are mostly confounded by the time of the day (see Methods section and reported abiotic conditions in S7 Fig). In this context, it was, therefore, difficult to statistically study the individual effects of time of the day and abiotic variables (temperature and humidity) on capture rate. We are confident that Sticky Pi could, scaled at the landscape level, with explicitly different microclimates, help address the interplay between abiotic variables and circadian processes.

In the last few years, we have seen applications of chronobiology to fields such as learning [40] and medicine [41]. We argue that chronobiological considerations could be equally important to biodiversity conservation and precision agriculture [42–44]. For example, plants' defences [45,46] and insecticide efficiency [47,48] may change during the day, implying that agricultural practices could be chronobiologically targeted. In addition, modern agriculture is increasingly relying on fine-scale pest monitoring and the use of naturally occurring biological pest control [49,50]. Studying insect community chronoecology could help predict the strength of interactions between a pest and its natural enemies or measure the temporal patterns of recruitment of beneficial natural enemies and pollinators. Monitoring insect behaviours at high temporal resolution is critical for both understanding, forecasting, and controlling emerging insect pests in agriculture and, more broadly, to comprehend how anthropogenic activities impact behaviour and biodiversity of insect populations.

## Methods

### Image processing

**Universal insect detector.** *Data*. We acquired a diverse collection of 483 Sticky Pi images by, first, setting stand-alone sticky cards in the Vancouver area for 1 to 2 weeks and taking images with a Sticky Pi afterwards, and, second, by systematically sampling images from the field experiments. The first set of "offline" images was physically augmented by taking photographs in variable conditions, which included illumination, presence of water droplets and thin dust particles. In order to generalise our algorithm, we also collected another "foreign" 171 images of sticky cards acquired with other devices. Among the foreign images, 140 were acquired by ourselves using the Raspberry Pi camera HQ (in 2021), and 31 were provided by the community (digital cameras and desktop scanner)—see Acknowledgments. We annotated images using Inkscape SVG editor, encoding annotations as SVG paths. The outline of each visible arthropod was drawn. The contours of 2 adjacent animals were allowed to overlap. We automatically discarded objects smaller than 30 px (i.e., <2 mm objects that are indiscernible in the images by manual annotators) or wider than 600 px (i.e., objects larger than 40 mm, which were generally artefacts since the vast majority of captured insects are smaller in our

study region). Partial insects were only considered if their head and thorax were both visible. This procedure resulted in a total of 33,556 segmented insects.

*Training.* To perform instance segmentation, we used Mask R-CNN [17]. In order to train the algorithm, images were pseudo-randomly split into a validation (25%, 96 images) and a training (75%, 387 images) set, based on their md5 checksum. In order to account for partial insects on the edge of the pictures all images were zero-padded with a 32 px margin. We performed augmentation on the training set as follow. First, random regions of 1024×1024 px were cropped in the padded images. Then, we applied, to each image the following: random rotation (0 to 360 degrees); random vertical and horizontal reflections; and alterations of saturation, brightness, contrast, and hue (uniform random in [0.9, 1.1]). We use the detectron2 implementation [51] of Mask R-CNN to perform the instance segmentation (insect versus background). We retrained a ResNet50 conv4 backbone, with conv5 head, which was pretrained on the COCO dataset, for 150,000 iterations (12 images per batch) with an initial learning rate of 0.002, decaying by $\gamma = 0.8$ every 10,000 iterations.

*Generalisation to large images.* The default standard dimension of Mask R-CNN inputs is 1024×1024 px. Our images being larger (2592×1944 px), we performed predictions on 12 1024×1024 tiles (in a 4×3 layout), which mitigates edge effects since tiles overlap sufficiently so that very large insects ($> 500$ px wide) would be complete in, at least, one tile. A candidate insect instance (defined as a polygon) $B$ was considered valid if and only if $J(A_i, B) < 0.5 \forall i$, where $A_i$ represents valid instances in neighbouring tiles, and $J$ is the Jaccard index.

**Siamese insect matcher.** *Matching function.* The goal of the SIM is to track insect instances through consecutive frames—given that insects may move, escape, be predated, get occluded, etc. The core of the algorithm is the matching function $M(m, n) \in [0, 1]$, between objects $m$ and $n$ detected by the Universal Insect Detector. This section describes how $M(m, n)$ is computed (see also S3A Fig for a visual explanation). In order to compute $M$, we opted for a mixture of visual similarity and explicit statistics such as differences in area and position between 2 instances.

For visual similarity, we reasoned that we could extract 2 variables. First, the naive similarity ($S$) is the similarity between the image of an insect in a given frame and the image of another (putatively the same) in a subsequent frame. We compute such similarity using a Siamese neural network. Second, an important information is whether an insect present in a given frame has actually moved away in the next frame. To assess such "delayed self-similarity" ($Q$), we can use the same network, as we are asking the same question. Indeed, intuitively, if we detect 2 similar insects in 2 consecutive frames ($S$), but when looking at the exact same place as the first insect, in the second image, we find a very high similarity, it suggest the original insect as, in fact, not moved.

Formally, given a pair of objects $m$, $n$, in images $\mathbf{X}^i$ and $\mathbf{X}^j$, we have the binary masks $A_m$ and $A_n$ of $m$ and $n$, respectively. We then use the same function $D$ to compute 2 similarity values $S(m, n)$ and $Q(m, n)$. With

$$S(m, n) = D(\mathbf{X}^i \cap A_m, \mathbf{X}^j \cap A_n),$$

i.e., the similarity between $m$ in its original frame, $i$, and $n$ in its original frame, $j$. And,

$$Q(m, n) = D(\mathbf{X}^i \cap A_m, \mathbf{X}^j \cap A_m),$$

i.e., the similarity between $m$ in its original frame, $i$, and $m$ projected in the frame $j$. Note, that all inputs are cropped to the bounding box of $A$, and scaled to 105×105 px. $D$ is a Siamese network as defined in [52] with the notable distinction that the output of our last convolutional layer has a dimension of 1024×1 (versus 4096×1, in the original work), for performance

reasons. In order to integrate the nonlinear relationships between the 2 resulting similarity values, $S(m, n)$ and $Q(m, n)$, as well as other descriptive variables, we used a custom, 4-layers, fully connected neural network, $H(I)$. The inputs are

$$I = \{S(m, n), Q(m, n), d(C(m), C(n)), |log(\mathcal{A}_m/\mathcal{A}_n)|, log(\Delta t + 1)\},$$

where $d$ is the Euclidean distance between the centroids $C$. $\mathcal{A}$ is the area of an object, and $\Delta t = t_j - t_i$. Our 4 layers have dimensions 5,4,3,1. We use a ReLU activation function after the first 2 layers and a sigmoid function at the output layer.

*Data*. In order to train the SIM core Matching function $M$, we first segmented image series from both berry field trials (see below) with the Universal Insect Detector to generate annotations (see above). We randomly sampled pairs of images from the same device, with the second image between 15 min and 12 h after the first one. We created a composite SVG image that contained a stack of the 2 images, and all annotations as paths. We then manually grouped (i.e., SVG groups) insects that were judged the same between the 2 frames. We generated 397 images this way, containing a total of 9,652 positive matches. Negative matches ($N = 200{,}728$) were defined as all possible nonpositive matches between the first and second images. Since the number of negatives was very large compared to the positive matches, we biased the proportion of negative matches to 0.5 by random sampling during training.

*Training*. We trained the SIM in 3 steps. First, we pretrained the Siamese similarity function $D$ by only considering the $S(m, n)$ branch of the network (i.e., apply the loss function on this value). Then we used the full network, but only updated the weights of the custom fully connected part $H(I)$. Last, we fine-tuned by training the entire network. For these 3 steps, we used Adaptive Moment Estimation with learning rates of $2 \times 10^{-5}$, 0.05, and $2 \times 10^{-5}$, for 500, 300, and 5,000 rounds, respectively. We used a learning rate decay of $\gamma = 1 - 10^{-3}$ between each round. Each round consisted of a batch of 128 pairs. We defined our loss function as binary cross-entropy.

*Tracking*. We then use our instance overall matching function ($M$) for tracking insects in 3 consecutive steps. We formulate this problem as the construction of a graph $G(V, E)$, with the insect instances in a given frame as vertices $V$, and connection to the same insect, in other frames, as edges $E$ (see also S3B Fig for a visual explanation). This graph is directed (through time), and each resulting (weakly) connected subgraph is an insect "tuboid" (i.e., insect instance). Importantly, each vertex can only have a maximum of one incoming and one outgoing edge. That is, given $v \in V$, $\deg^-(v) \leq 1$ and $\deg^+(v) \leq 1$. We build $G$ in 3 consecutive steps.

First, we consider all possible pairs of instances $m, n$ in pairs of frames $i,j$, with $j = i+1$ and compute $M(m, n)$. In other words, we match only in contiguous frames. Then, we define a unique edge from vertex $m$ as

$$e = \begin{cases} \emptyset \text{ if } M(m, n) < k \forall n \\ \{(m, \underset{n}{arg\ max}\ M(m, n))\} \text{ else} \end{cases}, \tag{1}$$

where $k = 0.5$ is a threshold on $M$. That is, we connect an instance to the highest match in the next frame, as long as the score is above 0.5. We obtain a draft network with candidate tuboids as disconnected subgraphs.

Second, we apply the same threshold (Eq. 1) and consider the pairs all pairs $m, n$, in frames $i, j$, where $\deg^+(m) = 0$, $\deg^-(n) = 0$, $j - i > 1$ and $t_j - t_i < 12h$. That is, we attempt to match the last frame of each tuboid to the first frame of tuboids starting after. We perform this operation recursively, always connecting the vertices with the highest overall matching score, and

restarting. We stop when no more vertices match. This process bridges tuboids when insects were temporarily undetected (e.g., occluded).

Finally, we define any 2 tuboids $P(E,V)$ and $Q(F,W)$ (i.e., disconnected subgraphs, with vertices $V$ and $W$, and edges $E$ and $F$) as conjoint if and only if $t_v \neq t_w \forall v,w$, and $min(t_v) \in [min(t_w), max(t_w)]$ or $min(t_w) \in [min(t_v), max(t_v)]$. That is, 2 tuboids are conjoint if and only if they overlap in time, but have no coincidental frames. We compute an average score between conjoint tuboids as

$$\bar{M}(P, Q) = \frac{1}{N}\sum_{v,w \in K} M(v, w),$$

where $K$ is the set of $N$ neighbouring pairs, in time:

$$K = \bigcup_v \{(v, \; \underset{w \forall t_v > t_w}{arg\; min}(t_v - t_w)), (v, \underset{w \forall t_v < t_w}{arg\; min}(t_w - t_v)\}$$

That is, the average matching score between all vertices and their immediately preceding and succeeding vertices in the other tuboid. We apply this procedure iteratively with a threshold $k = 0.25$, merging first the highest-scoring pair of tuboids. Finally, we eliminate disconnected subgraphs that do not have, at least, 4 vertices.

**Insect Tuboid classifier.** *Data*. We generated tuboids for both field trials (see below) using the SIM described above. We then visually identified and annotated a random sample of 4003 tuboids. Each tuboid was allocated a composite taxonomic label as type/order/family/genus/species. Type was either Background (not a complete insect), Insecta, or Ambiguous (segmentation or tracking error). It was not possible to identify insects at a consistent taxonomic depth. Therefore, we characterised tuboids at a variable depth (e.g., some tuboids are only Insecta/* while others are Insecta/Diptera/Drosophilidae/Drosophila/D. suzukii).

*Training*. In order to train the Insect Tuboid Classifier, we defined 18 flat classes (i.e., treated as discrete levels rather than hierarchical; see Fig 3). We then pseudo-randomly (based on the image md5 sum) allocated each tuboid to either the training or the validation data subset, ensuring an approximate $\frac{3}{4}$ to $\frac{1}{4}$, training to validation, ratio, per class. We excluded the 125 ambiguous annotations present, resulting in a total of 2,896 training and 982 validation tuboids. We initialised the weight of our network from a ResNet50 backbone, which had been pretrained on a subset of the COCO dataset. For our loss function, we used cross-entropy, and stochastic gradient descent as an optimizer. We set an initial learning rate of 0.002 with a decay $\gamma = 1 - 10^{-4}$ between each round and a momentum of 0.9. A round was a batch of 8 tuboids. Each individual image was augmented during training by adding random brightness, contrast, and saturation, randomly flipping along x and y axes and random rotation $[0, 360]°$. All individual images were scaled to 224×224 px. Batches of images we normalised during training (standard for ResNet). We trained our network for a total of 50,000 iterations.

**Implementation, data, and code availability.** We packaged the source code of the image processing as a python library: sticky-pi-ml (https://github.com/sticky-pi/sticky-pi-ml). Our work makes extensive use of scientific computing libraries OpenCV (Bradski, 2000), Numpy [53], PyTorch [54], sklearn [55], pandas [56], and networkx [57]. Neural network training was performed on the Compute Canada platform, using a NVidia 32G V100 GPU. The dataset, configuration files, and resulting models for the Universal Insect Detector, the SIM, and the Insect Tuboid Classifier are publicly available, under the creative commons license [58]. The underlying data for the relevant figures are publicly available [20].

## Laboratory experiments

In order to reproduce classic circadian experiments in an established model organism, we placed approximately 1500 $CO_2$-anesthetised *D. melanogaster* individuals in a 950 mL (16 oz) deli container, with 100 mL of agar (2%), sucrose (5%) and propionic acid (0.5%) medium. The top of this primary container was closed with a mosquito net, and a 3-mm hole was pierced on its side, 40mm from the bottom and initially blocked with a removable cap. Each cup was then placed in a large (25×25×50 cm) rectangular cage (secondary container), and all cages were held inside a temperature-controlled incubator. In the back of each cage, we placed a Sticky Pi device that had been modified to use infrared, instead of visible, light. In addition, we placed 100 mL of media in an open container inside each cage, so that escaping animals could freely feed. Flies were left at least 48h to entrain the light regime and recover from anaesthesia before the small aperture in their primary container was opened. The small diameter of the aperture meant that the escape rate, over a few days, was near stationary. The *D. melanogaster* population was a mixture of CantonS males and females from 3 to 5 days old, and the number of individuals was approximated by weighting animals (average fly weight = $8.4×10^{-4}$g). During the experiments, the temperature of the incubators was maintained at 25˚C and the relative humidity between 40% and 60%. All animals were entrained in a 12:12 h Light:Dark regime. Flies were kindly given by Mike Gordon (University of British Columbia). One experimental replicate (device × week) was lost due to a sticky card malfunction.

## Field experiments

In order to test the ability of the Sticky Pi device to capture the daily activity patterns of multiple species of free-living insects, we deployed 10 prototype devices on an experimental farm site in Agassiz, British Columbia, Canada (GPS: 49.2442, -121.7583) from June 24 to September 30, 2020. The experiments were done in 2 plots of berry plants, raspberries, and blackberries, which mature and decline during early and late summer, respectively. Neither plot was sprayed with pesticides at any point during the experiments. Temperature and humidity data extracted from the DHT22 sensors of the Sticky Pis are reported in S7 Fig. None of the species identified during this study were protected species.

**Blackberry field.**   The blackberry (*Rubus fruticosis* var. "Triple Crow"') plot was made up of 5 rows, each of which was approximately 60 metres long. Each row had wooden posts (approximately 1.6 m high) spaced approximately 8.5 m apart, along which 2 metal "fruiting wires" were run at 2 different heights (upper wire: 1.4 m; lower wire: 0.4 m). Two traps, 1 baited with apple cider vinegar and 1 unbaited, were set up on 2 randomly selected wooden posts within each of the 5 rows, with the position of baited and unbaited traps (relative to the orientation of the field) alternated among rows. A plastic cylindrical container (diameter: 10.6 cm; height: 13.4 cm) with 2 holes cut in the side (approximately 3×6 cm) and fine mesh (knee-high nylon pantyhose) stretched over the top, containing approximately 200 mL of store-bought apple cider vinegar was hung directly under baited traps (S5 Fig). No such container was hung under unbaited traps. Vinegar in the containers hung under baited traps was replaced weekly. Traps were affixed to the wooden posts at the height of the upper fruiting wire so that they faced southwards. Trapping locations did not change over the course of the experiment, which began approximately 2 weeks after the beginning of blackberry fruit ripening (August 12, 2020) and ended when fruit development had mostly concluded (September 30, 2020). Sticky cards were replaced once weekly, and photographs were offloaded from traps every 1 to 2 weeks. Approximately 15 trap days of data were lost during the experiment due to battery malfunctions. Overall, 475 trap days over 70 replicates (device × week), remained (i.e., 96.9%).

**Raspberry field.** A total of 10 Sticky Pi devices were set up in a raspberry (*Rubus idaeus* var. "Rudi") plot with 6 rows, each of which was approximately 50 m long. Each row had wooden posts (approximately 1.6 m high) spaced 10 m apart, along which 2 metal "fruiting wires" were run at 2 different heights (upper wire: 1.4 m; lower wire: 0.4 m) to support plants. Two traps were set up on a randomly selected wooden post within each of 5 randomly selected rows. At each location, to capture any potential fine-scale spatial variation in insect communities, traps were affixed to the wooden posts at 2 different heights, at the same levels as the upper and lower fruiting wires. Traps were oriented southwards. Trapping locations within the field did not change over the course of the experiment, which began approximately 1 week after the beginning of raspberry fruit ripening (June 24, 2020) and ended after fruiting had concluded (July 29, 2020). Sticky cards were replaced once weekly, and photographs were offloaded from traps every 1–2 weeks. Some data (approximately 9 days, from 3 replicates) were lost due to battery malfunctions. Overall, 271 trap days over 40 replicates (device × week), remained (i.e., 96.8%).

**DNA barcoding.** In order to confirm the taxonomy of visually identified insects, we recovered specimens from the sticky cards after the trials to analyse their DNA and inform visual labelling. We targeted the molecular sequence of the cytochrome c oxidase subunit I (CO1). The overall DNA barcoding workflow follows established protocols [59] with minor modifications. Briefly, the genomic DNA of insect specimens was extracted with the QIAamp Fast DNA Stool Mini Kit (QIAGEN, Germantown, Maryland, USA) according to the manufacturer's instructions. The resulting DNA samples were then subjected to concentration measurement by a NanoDrop One/OneC Microvolume UV-Vis Spectrophotometer (Thermo Fisher Scientific, Waltham, Massachusetts, USA) and then normalised to a final concentration of 50 ng/$\mu$l. Next, depending on the identity of the specimen, the following primer pairs were selected for CO1 amplification: C_LepFolF/C_LepFolR [60] and MHemF/LepR1 [61]. Amplification of the CO1 barcode region was conducted using Phusion High-Fidelity DNA Polymerase (New England Biolabs, Ipswich, Massachusetts, USA) with the following 25 $\mu$l reaction recipe: 16.55 $\mu$l ddH2O, 5 $\mu$l 5HF PCR buffer, 2 $\mu$l 2.5 mM dNTP, 0.6 $\mu$l of each primer (20 $\mu$M), 0.25 $\mu$l Phusion polymerase, and finally 2 $\mu$l DNA template. All PCR programmes were set up as the following: 95˚C for 2 min; 5 cycles at 95˚C for 40 s, 4˚C for 40 s, and 72˚C for 1 min; then 44 cycles at 95˚C for 40 s, 51˚C for 40 s, and 72˚C for 1 min; and a final extension step at 72˚C for 10 min. PCR products were then subjected to gel electrophoresis and then purified with EZ-10 Spin Column DNA Gel Extraction Kit (Bio Basic, Markham, Ontario, Canada). After Sanger sequencing, a Phred score cutoff of 20 was applied to filter out poor-quality sequencing reads. The barcode index number (BIN) of each specimen was determined based on 2% or greater sequence divergence applying the species identification algorithm available on the Barcode of Life Data Systems (BOLD) version 4 [62]. Barcode sequences were deposited in GenBank (accession number SUB11480448). We also took several high-quality images of each specimen before DNA extraction and embedded them in a single table (S2 Table) to cross-reference morphology and DNA sequences [63].

## Statistics and data analysis

**Laboratory.** Images from the laboratory trials were processed using a preliminary version of the Universal Insect Detector on independent frames—i.e., without subsequent steps. This resulted in a raw number of detected insects on each independent frame. In order to filter out high-frequency noise in the number of insects, we applied a running median filter (k = 5) on the raw data. Furthermore, we filtered the instantaneous capture rate ($dN/dt$) with a uniform linear filter (k = 5). These 2 operations act as a low-pass frequency filter, with an approximate span of 100 min.

**Warped Zeitgeber time.** Zeitgeber time (ZT) is conventionally used to express the time as given by the environment (typically light, but also temperature, etc.). By convention, ZT is expressed in hours, between 0 and 24, as the duration since the onset of the day (i.e., sunrise = ZT0). Using ZT is very convenient when reporting and comparing experiments in controlled conditions. However, ZT only defines a lower bound (ZT0) and is therefore difficult to apply when day length differs (which is typical over multiple days, under natural conditions, especially at high and low latitudes). In order to express time relatively to both sunrise and sunset, we applied a simple linear transformation of ZT to WZT, $W(z)$.

Like ZT, we set WZT to be 0 at sunrise, but to always be $\frac{1}{2}$ day at sunset and to scale linearly in between. Incidentally, WZT is $\frac{1}{4}$ day and $\frac{3}{4}$ day at solar noon and at solar midnight, respectively. Formally, we express WZT as a function of ZT with

$$W(z) = \begin{cases} az & \text{, if } z \leq d \\ a'z + b' & \text{, otherwise} \end{cases},$$

where, $a$, $a'$ and $b'$ are constants, $d$ is the day length, as a day fraction. $z \in [0,1)$ is ZT and can be computed with $z = t - s \bmod 1$, where $t$ is the absolute time and $s$, the time of the sunrise. Since WZT is 1 when ZT is 1, we have $W(1) = a'1 + b' = 1$ Also, WZT is $\frac{1}{2}$ at sunset: $W(d) = ad = a'd + b' = \frac{1}{2}$, Therefore, $a = \frac{1}{2d}$, $a' = \frac{1}{2(1-d)}$ and $b' = 1 - a'$.

**Multidimensional scaling.** We derived the distance $d$ between 2 populations, $x$ and $y$ from the Pearson correlation coefficient $r$, as $d_{xy} = \frac{1 - r_{xy}}{2}$. In order to assess the sensitivity of our analysis, we computed 500 bootstrap replicates of the original data by random resampling of the capture instances, with replacement, independently for each taxon. We computed one distance matrix and MDS for each bootstrap replicate and combined the MDS solutions [64]. The 95% confidence ellipses were drawn assuming a bivariate t-distribution.

**Implementation and code availability.** Statistical analysis and visualisation were performed in R 4.0 [65], with the primary use of packages, smacof [66], data.table [67], mgcv [68], maptools [69], ggplot2 [70], and rethomics [71]. The source code to generate the figures is available at https://github.com/sticky-pi/sticky-pi-manuscript.

## Supporting information

**S1 Fig. The Sticky Pi platform.** Sticky Pi devices acquire images that are retrieved using a data harvester—based on another Raspberry Pi. The data from the harvesters are then incrementally uploaded to a centralised, per-laboratory, database. Images are then automatically preprocessed (the Universal Insect Detector is applied). Users and maintainers can visualise data in real time using our Rshiny web application. The remote API is secured behind an Nginx server, and the images are saved on an S3 server. All components of the server are deployed as individual interacting Docker containers. API documentation and source code are available on https://doc.sticky-pi.com/web-server.html. API, Application Programming Interface. (EPS)

**S2 Fig. Validation of the Universal Insect Detector.** The Universal Insect Detector performs instance segmentation, using Mask R-CNN, on all images in order to detect insects versus background. **(A)** Representative image region from the validation dataset. **(B)** Manual annotation of A. **(C)** Automatic segmentation of A. Coloured arrows show qualitative differences between human label (B) and our algorithm: either false positives or false negatives, in blue and red, respectively. Note that often, samples are degraded and ambiguous, even for trained annotators. **(D)** Recall as a function of insect size, showing our method is more sensitive to

larger insects. **(E)** Precision as a function of insect size. Precision is overall stationary. The panels on top of D and E show the marginal distribution of insect areas as a histogram. Both curves on D and E are Generalised Additive Models fitted with binomial response variables. Ribbons are the models' standard errors. The validation dataset contains a total of 8,574 insects. The underlying data for this figure can be found on figshare [20].
(EPS)

**S3 Fig. Description of the SIM. (A)** The matching metric in the SIM is based on a Siamese network (see Methods section). **(B)** The resulting score, *M*, is used in 3 steps to connect insect instances between frames. The algorithm results in a series of tuboids, which are representations of single insects through time. SIM, Siamese Insect Matcher.
(EPS)

**S4 Fig. Temporal niches of insect taxa in a raspberry field community.** Complementary data to Fig 6, showing all predicted taxa. Full species names are in the legend of Fig 3 and in the result section. The low relative frequency of *Drosophila suzukii* in this unbaited trial and visual inspection suggest male *D. suzukii* are false positives. Other drosophilid-like flies appear to be unknown small diurnal Diptera. The underlying data for this figure can be found on figshare [20].
(EPS)

**S5 Fig. Baited sticky pi.** Sticky pi device (top) with an olfactory bait (bottom). The bait consists of a container holding 200 mL of apple cider vinegar protected behind a thin mesh. Apple cider vinegar was replaced weekly during trap maintenance.
(TIFF)

**S6 Fig. Capture rates are not impacted by previously captured insects. (A)** Cumulative number of insect over time, for each trap (black line). No noticeable reduction of trapping rate occur, even when large number of insects (>100) are captured. **(B)** Proportion of insect of a given taxa captured in the first 3 days of a week as a function of the final total number of all insects. The red dotted line indicate the null hypothesis: Half of the insects are captured in the first half of each experiment, and total number of captured insects does not affect capture rate of a given taxa. The blue lines, and their error bars are individual linear models, none of which show a significant slope or intercept, *t* tests on model coefficients.
(EPS)

**S7 Fig. Environmental conditions during both experiments.** Temperature and relative humidity throughout field experiment, as recorded by Sticky Pi's sensors (DHT 22). **(A)** The experiment in the raspberry field. **(B)** The experiment in the blackberry field. Black and white rectangles, below and above each plot show days (between sunrise and sunset) and night (between sunset and sunrise) in white and black, respectively. Sun position is inferred through the maptools package [69]. The average between-device standard deviations for hourly temperature and humidity reads were, respectively, 0.68˚C and 3.95% for the raspberry field (A) and 0.63˚C and 3.4% for the blackberry experiment (B).
(EPS)

**S1 Table. Description of the 18 taxonomical labels.** We selected 18 taxa as discrete labels based on both visual examination and DNA barcoding evidence. This table describes the selected groups. Data on figshare.
(XLSX)

**S2 Table. Representative insect specimen used for DNA barcoding.** Data are compiled as an excel spreadsheet with embedded images of the specimens of interest alongside the DNA

sequence of their CO1. their DNA-inferred taxonomy and, when relevant, some additional notes. The printed labels in the images are 5×1 mm wide. The column "label_in_article corresponds to the visually distinct group to which insects were allocated. Data on figshare. CO1, cytochrome c oxidase subunit I.
(XLSX)

**S3 Table. Confusion matrix for the Insect Tuboid Classifier.** Detailed confusion matrix for the 982 tuboids of the validation set. Rows and columns indicate ground truth and predicted labels, respectively. Abbreviated labels are *Macropsis fuscula* (3), *Drosophila suzukii* males (4), drosophilids that are not male *D. suzukii* (5), *Anthonomus rubi* (11), *Psyllobora vigintimaculata* (12), Coleoptera that do not belong to any above subgroup (14) and *Lasioglossum laevissimum* (16). Data on figshare.
(XLS)

**S1 Video. The Sticky Pi web application.** Web interface of the Sticky Pi cloud platform. Users can log in and select a data range and devices of interest. Then an interactive plot shows environmental conditions over time for each device. Hovering on graphs shows a preview of the images at that time. Users can click on a specific time point to create a pop-up slideshow with details about the image as well as preprocessing results (number and position insects). Data on figshare.
(MP4)

**S2 Video. Typical image series.** Video showing 1 week of data at 10 frames per second. Images are sampled approximately every 20 minutes. Note the variation of lighting, transient occlusions, and insects escaping or degrading. Data on figshare.
(MP4)

**S3 Video. Output of the SIM.** Each rectangle is a bounding box of an inferred insect instance. Data on figshare. SIM, Siamese Insect Matcher.
(MP4)

**S4 Video. Predation of trapped insects by gastropods.** Video showing the extent of slug predation on trapped insects in our dataset. Each rectangle is a bounding box of an inferred insect instance (i.e., tuboid). Data on figshare.
(MP4)

**S5 Video. *Anthonomus rubi* escaping a sticky trap.** Multiple individual strawberry blossom weevils (*A. rubi*) impact the trap, but the majority rapidly manage to escape. *A. rubi* is an emergent invasive pest in North America. Data on figshare.
(MP4)

## Acknowledgments

We thank all members of the Plant-Insect Ecology and Evolution Laboratory (UBC) and the Insect Biocontrol laboratory (AAFC) for their help. In particular, Warren Wong (UBC/ AAFC), Matt Tsuruda (UBC), Dr. Pierre Girod (UBC), Sara Ng (UBC), Yuma Baker (UBC), Jade Sherwood (AAFC/University of the Fraser Valley), and Jenny Zhang (UBC/AAFC) for helping with design decisions, tedious image annotations, the literature search, and the design of the lab experiments. We thank Dr. Mike Gordon (UBC) for providing us with *Drosophila melanogaster* CS flies. We are very grateful to Dr. Mark Johnson (UBC), Dr. Esteban Beckwith (Imperial College London), Dr. Alice French (Imperial College London), Luis García Rodríguez (Universität Münster), Mary Westwood (University of Edinburgh), and Dr. Lucia Prieto

(Francis Crick Institute) for their very constructive advice and discussions on various aspects of the project. We thank Devika Vishwanath, Samia Siddique Sama, and Priyansh Malik, students of the Engineering Physics program (UBC) as well as their mentors for their ongoing work on the next version of the Sticky Pi. The Compute Canada team provided remarkable support and tools for this project. We are very thankful to Dr. Chen Keasar, Dr. Dan Rustia [72], and Kim Bjerge [25] for making their images available or for providing "foreign" images to extend the scope of the Universal insect detector. We acknowledge that some of the research described herein occurred on the traditional, ancestral, and unceded xwmkwým Musqueam territory and on the traditional lands of the Sto:lo people.

## Author Contributions

**Conceptualization:** Quentin Geissmann, Paul K. Abram, Cara H. Haney, Juli Carrillo.

**Data curation:** Quentin Geissmann.

**Formal analysis:** Quentin Geissmann.

**Funding acquisition:** Quentin Geissmann, Cara H. Haney, Juli Carrillo.

**Investigation:** Quentin Geissmann, Paul K. Abram, Di Wu.

**Methodology:** Quentin Geissmann, Paul K. Abram, Di Wu.

**Project administration:** Quentin Geissmann, Juli Carrillo.

**Resources:** Quentin Geissmann, Paul K. Abram, Di Wu, Juli Carrillo.

**Software:** Quentin Geissmann.

**Supervision:** Cara H. Haney, Juli Carrillo.

**Validation:** Quentin Geissmann, Paul K. Abram.

**Visualization:** Quentin Geissmann.

**Writing – original draft:** Quentin Geissmann, Paul K. Abram, Di Wu, Cara H. Haney, Juli Carrillo.

**Writing – review & editing:** Quentin Geissmann, Paul K. Abram, Di Wu, Cara H. Haney, Juli Carrillo.

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
