## [Editor Report · Decision Letter 0]

21 Oct 2021

Dear Dr Geissmann, 

Thank you for submitting your manuscript entitled "Sticky Pi, a high-frequency smart trap to study insect circadian activity under natural conditions" for consideration as a Methods and Resources by PLOS Biology.

Your manuscript has now been evaluated by the PLOS Biology editorial staff as well as by an academic editor with relevant expertise and I am writing to let you know that we would like to send your submission out for external peer review.

Once your full submission is complete, your paper will undergo a series of checks in preparation for peer review. Once your manuscript has passed the checks it will be sent out for review. 

If your manuscript has been previously reviewed at another journal, PLOS Biology is willing to work with those reviews in order to avoid re-starting the process. Submission of the previous reviews is entirely optional and our ability to use them effectively will depend on the willingness of the previous journal to confirm the content of the reports and share the reviewer identities. Please note that we reserve the right to invite additional reviewers if we consider that additional/independent reviewers are needed, although we aim to avoid this as far as possible. In our experience, working with previous reviews does save time. 

If you would like to send your previous reviewer reports to us, please specify this in the cover letter, mentioning the name of the previous journal and the manuscript ID the study was given, and include a point-by-point response to reviewers that details how you have or plan to address the reviewers' concerns. Please contact me at the email that can be found below my signature if you have questions. 

Please re-submit your manuscript within two working days, i.e. by Oct 25 2021 11:59PM.

Kind regards,

Lucas

Lucas Smith

Associate Editor

PLOS Biology

lsmith@plos.org

---

## [Decision Letter · Decision Letter 1]

1 Dec 2021

Dear Dr Geissmann,

Thank you for submitting your manuscript "Sticky Pi, a high-frequency smart trap to study insect circadian activity under natural conditions" for consideration as a Methods and Resources article at PLOS Biology. Your manuscript has been evaluated by the PLOS Biology editors, an Academic Editor with relevant expertise, and by several independent reviewers.

The reviews of your manuscript are appended below. As you will see from their comments, the reviewers are generally positive about the study and its potential usefulness for the field. However, they have also raised a number of concerns which limit the broader impact of the study and which will need to be thoroughly addressed before we can consider your manuscript for publication. The reviewers have particularly highlighted that the generalizability of the deep learning pipeline has not been sufficiently demonstrated, and that the method has not been sufficiently validated and benchmarked in the lab or against existing trapping approaches in the field. Furthermore, the reviewers have highlighted a number of important limitations which require additional discussion and/or analyses to exclude.

In light of the reviews, we will not be able to accept the current version of the manuscript, but we would welcome re-submission of a much-revised version that takes into account the reviewers' comments. We cannot make any decision about publication until we have seen the revised manuscript and your response to the reviewers' comments. Your revised manuscript is also likely to be sent for further evaluation by the reviewers. 

We expect to receive your revised manuscript within 3 months. 

**IMPORTANT - SUBMITTING YOUR REVISION**

*Re-submission Checklist*

*Published Peer Review*

*PLOS Data Policy*

*Blot and Gel Data Policy*

Sincerely,

Lucas Smith

Associate Editor

PLOS Biology

lsmith@plos.org

REVIEWS:

Reviewer #1, Eamonn Keogh: Very nice paper. Well motivated, well explain, great experimental design, high reproducibility, good illustrations, open source etc.

However..

I think this is the wrong solution.

1) You are getting counts every 20 minutes, but that will obscure very finely timed behaviors, consider fig 5 of [a]. It would be easy to get every second.

2) The proposed system is computationally very heavyweight, but simpler hardware and software can do a better job [a].

3) The system is destructive, you can only count dead insects, but it would be sometimes useful to count live insects, without killing them (for A|B tests etc)

4) At some point you have sticky trap situation, either with dead insects, or with dust. At that point you can no longer say anything. Using [a] means you can always count. 

I do worry about the generalizability of deep learning. It is difficult not to have them overfit. How robust are they to type of camera, color of sticky trap, distance from lens to trap, ambient light etc

-

The videos are very helpful.

I hope when deployed, you put a grill of some kind around the trap, to prevent birds and bats being caught in the sticky traps. In some places, you will also need to exclude geckos and lizards (I have found Teflon sheets around the support poles works very well)

[a] Yanping Chen , Adena Why, Gustavo Batista, Agenor Mafra-Neto, Eamonn Keogh. Flying Insect Classification with Inexpensive Sensors. Journal of Insect Behavior 2014

http://alumni.cs.ucr.edu/~ychen053/InsectBehaviour_059.pdf

Reviewer #2: In this methods article, Geissmann and colleagues present a novel tool to perform insect surveys in the field. The Sticky Pi tool consists of one (or more) 'recorder' devices which consist of a camera, weather station and a sticky insect trap. The trap is photographed at a set time interval. The data is then periodically transferred to a server using a 'data harvester', another device that prospective users need to build. 

To analyze the large amount of data the authors employ a three step analysis: first animals found on the trap are separated from the background, then individuals are tracked over time and finally each individual is assigned to a species. The authors collect and analyze a total of 3 datasets to support the functionality and usefulness of the tool. The results of the analysis are as expected (but appropriate statistics needs to be added, see below) indicating that Sticky Pi performs as claimed.

This manuscript is submitted as a methods paper. As such the reader should expect that enough information is provided to build the setup and run experiments as described in the manuscript. The paper is accompanied with a beautiful website which explains how one can build the hardware components. However, the generalizability of the machine learning based analysis pipeline is far less clear. 

Overall, the authors present an important and exciting tool to automate insect surveys, but the manuscript would greatly benefit from additional information as well as clarifications related to the analysis pipeline which is an integral part of the Sticky Pi tool. Moreover, it would be important to further benchmark the performances of the species identification (see below).

Major concerns:

A main concern with the information described in the manuscript is the applicability of the analysis pipeline by other users. Could the authors clarify the following points?

1) Can the Universal Insect Detector be used with different datasets? If yes, did you use the identical network for the 3 datasets presented in the manuscript? If you have other data that can not be used with the Universal Insect Detector, how would you go about training the network? 

2) Can the Siamese Insect Matcher be used with different datasets than those presented in the manuscript? Did you use the identical network for the 3 datasets in the manuscripts? If not, how you would go about training the network with new data? 

3) Obviously, the Insect Tuboid Classifier needs to be adapted to different data. Is it realistic to expect to just re-train the network with one's own classifiers? Or do you foresee the need to adapt the ResNet50 architecture itself? Please clearly outline steps someone would need to take to get optimal results.

4) For each of the networks: How does one go about labelling the data wherever it is necessary? Do you provide an interface to perform labelling or do you recommend other software? Please make sure the labelled data can be feed into your analysis pipeline without problems. 

The authors need to show that their analysis pipeline will be accessible and useable for other researchers.

Another concern is the sensitivity of the species identification algorithm. The recall rate reported in Supp. Fig . S2 is 70% when all objects were included — this rate is modest. An increase to 80% is obtained when the smallest insects were excluded from the analysis. While the size filter might be unavoidable, the sensitivity of a 80% recall rate suggests that the abundance of some species might be underestimated. For this reason, it would be important to benchmark the performance of the classifiers in lab conditions. For instance, the authors could prepare a mixed population of D. melanogaster, D. suzukii and a handful of other insects. This population could be placed in a cage with a Sticky Pi device (as was done for Figure 4). In these controlled conditions with known species, the authors should be able to assess the performances of the automated identification. This analysis would be useful to validate the results of the tracking of D. suzukii versus other Drosophilidae presented in Figure 5.

Figure 4:

The authors use the data in this figure to conclude that their tool can capture circadian rhythm. Although the data look promising, it is necessary to add appropriate statistics to support statements made in the text. This should also address why the LD control in C and D are different (by eye): You have more captures in D both in LD and DD and a generally lower baseline of morning capture in C and no capture in LL.

Why did you remove high frequency noise in this dataset but not the others? (Line 449)

Figure 5: 

In this well-designed figure, the authors compare male D. suzukii capture with wasp capture. They conclude that: "Our results corroborate a distinctive crepuscular activity pattern for male D. suzukii and other putative drosophilids. In contrast, Figitidae wasps were exclusively diurnal." Please provide appropriate statistics to support this statement. 

In addition, the authors state that baiting increases the number of male of D. suzukii being captured. In addition to the stastistics, please provide an appropriate plot showing these results. 

Minor issues: 

General:

What are the advantages of using white light for illumination? You showed in Figure 4 that IR light can be used. Please discuss this point.

Website:

Please update the BOM: for example the link for the GPS module says 'currently unavailable' which would make it impossible to build the data harvester.

Link to 3D printer files doesn't work as expected: Using the link to Onshape, the reviewer only gets one file 6-Pin Shrouded Header (IDC) Right Angle. 

Link to OS for data harvester doesn't work.

Figure 1:

Please clarify whether the sticky pad it must be transparent since the LEDs placed behind the pad? Are all sticky pads/cards transparent or did you have to select a particular brand?

Consider adding a picture or illustration of the data harvester.

Figure 3:

A) Please clearly indicate that input is from Siamese network (not images).

B) Please consider using a heatmap as the presentation representation is very hard to read.

C) In legend, F1 score is not explained.

D) Please consider adding scalebars - if the reader is not mistaken, everything has been rescaled. Therefore, out-of-focus images (e.g., 2) are of insects which are smaller than 'sharper' (e.g. 7) ones? This could be made more explicit in the legend.

Figure 4:

Please rearrange the legend (LD, LL and DD at top): Currently the LL is in the center while all data is presented in a left (LD/LL) and right (LD and DD) column.

If the reader understands correctly, y-axis in 4C and D is the same as in Fig5 A/B and Fig 6A. Please label consistently throughout the manuscript. 

As much as possible, please normalize the size of indivudal figures. At the moment, Figure 4 seems to be shrunk compared to Figure 5. This is obvious when comparing the labels of the plots. 

Figure 6:

Missing x axis label under 'Culicidae'. 

Figure S3:

Regarding the calculation of M(m,n): In methods (line 275), you write that "Given a pair of objects m, n, in images Xi and Xj, we have the binary masks Am and An of m and n, respectively. We then use the same function D to compute two similarity values S(m, n) and Q(m, n)." In your Figure S3A you have a total of 3 inputs with the lower looking like the difference between the binary mask of the two? Please elaborate in text why you chose to show the 3 inputs. 

Following the Siamese network you use a "custom, four-layers, fully connected neural network" (line 275). Why did you choose that particular architecture and those inputs? Do you expect this network to perform well on different datasets? 

Discussion:

It would seem appropriate to discuss the different approach Sticky Pi takes (decentralization of computation) compared to other published tools like this https://doi.org/10.3390/s21020343.

Cover page: There appears to be typos in affiliation #3.

Reviewer #3: The authors present a 'High-frequency monitoring tool to study insect circadian biology in the field' with applications in 'biodiversity monitoring, pest control, phenology, behavioural ecology, ecophysiology'. Their methodology represents an interesting advance in temporal monitoring of insects (and other captured species) in the field. The new methodology is tested in laboratory and field conditions in both baited and unbaited scenarios. While the computational limitations of the methodology receive scrutiny and validation, the manuscript is less sophisticated in discussing potential limitations in the interpretation of the collected data. The temperature and humidity conditions in the field were presumably monitored by the devices, but they were not reported as far as I can tell. This is relevant as diurnal insect behaviour changes strongly with environmental temperature patterns. Formally, what is measured is not the temporal field activity of insects per se, but rather the pattern of their trapping on the sticky surface of the device used with or without a bait being present. This adds further limitations that are worth acknowledging explicitly in addition to the limited taxonomic resolution that was mentioned. Trapping patterns will depend on the presence of bait (see notable difference for D. suzukii), competition with attractants present in the field, and further interactions with biotic and abiotic factors. In the case of the 'Sticky Pi' device one would also want to explore possible unintended impacts of light flashes associated with image capture as well as the impact of prior captures on the trap on subsequent ones in association with emitted attractant/repellent signals. It is also not clear whether the trap in its current form was comparatively tested against any other designs. A side-by-side validation relative to alternative trapping methods in the same field conditions would provide much clearer insight into 'Sticky Pi''s usefulness in a particular context than the notion that D. suzukii male behaviour appeared to be 'crepuscular' under field conditions with unspecified environmental profiles. In summary, while this study should be welcomed as a technical advance that may have a number of useful applications, it lacks rigorous field-based validation as well as a discussion of potential pitfalls associated with linking recorded data to field behaviours.

---

## [Editor Report · Decision Letter 2]

10 May 2022

Dear Dr Geissmann,

Thank you for your patience while we considered your revised manuscript "Sticky Pi, a high-frequency smart trap to study insect circadian activity under natural conditions" for publication as a Methods and Resources at PLOS Biology. This revised version of your manuscript has been evaluated by the PLOS Biology editors and the Academic Editor. We are satisfied by the changes made in the revision, which we think has done a good job of addressing the reviewer concerns. 

However, before we can accept this manuscript for publication, we will need you to revise your manuscript to satisfactorily address the following editorial, data, and policy-related requests:

1) After some discussion within the team, we think the title should be edited for clarity. If you agree, we suggest it be changed to something like ""Sticky Pi is a high-frequency smart trap that enables study of insect circadian activity under natural conditions" 

2) Thank you for providing supplementary data tables and videos on Figshare. In order to be compliant with our Data Policy, which requires that all data be made available without restriction, we request that you also provide, as a supplementary file or deposition in a publicly available repository, the underlying data used to generate each figure. 

Note that we do not require all raw data. Rather, we ask that all individual quantitative observations that underlie the data summarized in the figures and results of your paper be made available. 

Please ensure that you provide the individual numerical values that underlie the summary data displayed in the following figure panels as they are essential for readers to assess your analysis and to reproduce it:

Fig 4A-F; 5A-B; 6A-B; Fig S2 D-E; Fig S4

**Please also ensure that figure legends in your manuscript include information on where the underlying data can be found, and ensure your supplemental data file/s has a legend.

**Please ensure that your Data Statement in the submission system accurately describes where your data can be found.

For more details on our data availability policy, see here: http://journals.plos.org/plosbiology/s/data-availability

3) In order to comply with PLOS' ethics policy regarding field and observational studies (https://journals.plos.org/plosone/s/submission-guidelines#loc-observational-and-field-studies), we request that you add the following information to your methods section: 

-Please provide details on permits and approvals obtained for the field experiments performed here, including the full name of the authority that approved the study and the approval numbers; if none were required, please explain why

- Please indicate whether the land accessed is privately owned or protected

- Please indicate whether any protected species were sampled

4) Please provide scale bars for images in Figure 2 and Figure S2. 

As you address these items, please also take this last chance to review your reference list to ensure that it is complete and correct. If you have cited papers that have been retracted, please include the rationale for doing so in the manuscript text, or remove these references and replace them with relevant current references. Any changes to the reference list should be mentioned in the cover letter that accompanies your revised manuscript.

We expect to receive your revised manuscript within two weeks. 

*Published Peer Review History*

*Press*

Sincerely,

Lucas

Lucas Smith, Ph.D.,

Associate Editor,

lsmith@plos.org,

PLOS Biology

---

## [Editor Report · Decision Letter 3]

26 May 2022

Dear Dr Geissmann,

Thank you for the submission of your revised Methods and Resources "Sticky Pi is a high-frequency smart trap that enables the study of insect circadian activity under natural conditions" for publication in PLOS Biology. On behalf of my colleagues and the Academic Editor, Tom Baden, I am pleased to say that we can in principle accept your manuscript for publication, provided you address any remaining formatting and reporting issues. These will be detailed in an email you should receive within 2-3 business days from our colleagues in the journal operations team; no action is required from you until then. Please note that we will not be able to formally accept your manuscript and schedule it for publication until you have completed any requested changes.

**IMPORTANT: As you address these last formatting requests, to come, we also ask that you add part of your response to our previous editorial requests to your manuscript. Specifically, we ask that you indicate in the methods section that none of the species identified in your traps were protected species. 

PRESS

Sincerely, 

Lucas Smith, Ph.D. 

Associate Editor

PLOS Biology

lsmith@plos.org